# Herring Milt and Herring Milt Protein Hydrolysate Are Equally Effective in Improving Insulin Sensitivity and Pancreatic Beta-Cell Function in Diet-Induced Obese- and Insulin-Resistant Mice

**DOI:** 10.3390/md18120635

**Published:** 2020-12-11

**Authors:** Yanwen Wang, Sandhya Nair, Jacques Gagnon

**Affiliations:** 1Aquatic and Crop Resource Development Research Center, National Research Council of Canada, Charlottetown, PE C1A 4P3, Canada; sandhya.nair@nrc-cnrc.gc.ca; 2Department of Biomedical Sciences, University of Prince Edward Island, Charlottetown, PE C1A 4P3, Canada; 3VALORĒS Research Institute, Shippagan, NB E8S 1J2, Canada; 4Department of Sciences, Shippagan Campus, University of Moncton, Shippagan, NB E8S 1P6, Canada

**Keywords:** blood glucose, herring milt dry powder, herring milt protein hydrolysate, high-fat diet, HOMA-β, HOMA-IR, insulin resistance, mice, oral glucose tolerance, QUICKI

## Abstract

Although genetic predisposition influences the onset and progression of insulin resistance and diabetes, dietary nutrients are critical. In general, protein is beneficial relative to carbohydrate and fat but dependent on protein source. Our recent study demonstrated that 70% replacement of dietary casein protein with the equivalent quantity of protein derived from herring milt protein hydrolysate (HMPH; herring milt with proteins being enzymatically hydrolyzed) significantly improved insulin resistance and glucose homeostasis in high-fat diet-induced obese mice. As production of protein hydrolysate increases the cost of the product, it is important to determine whether a simply dried and ground herring milt product possesses similar benefits. Therefore, the current study was conducted to investigate the effect of herring milt dry powder (HMDP) on glucose control and the associated metabolic phenotypes and further to compare its efficacy with HMPH. Male C57BL/6J mice on a high-fat diet for 7 weeks were randomized based on body weight and blood glucose into three groups. One group continued on the high-fat diet and was used as the insulin-resistant/diabetic control and the other two groups were given the high-fat diet modified to have 70% of casein protein being replaced with the same amount of protein from HMDP or HMPH. A group of mice on a low-fat diet all the time was used as the normal control. The results demonstrated that mice on the high-fat diet increased weight gain and showed higher blood concentrations of glucose, insulin, and leptin, as well as impaired glucose tolerance and pancreatic β-cell function relative to those on the normal control diet. In comparison with the high-fat diet, the replacement of 70% dietary casein protein with the same amount of HMDP or HMPH protein decreased weight gain and significantly improved the aforementioned biomarkers, insulin sensitivity or resistance, and β-cell function. The HMDP and HMPH showed similar effects on every parameter except blood lipids where HMDP decreased total cholesterol and non-HDL-cholesterol levels while the effect of HMPH was not significant. The results demonstrate that substituting 70% of dietary casein protein with the equivalent amount of HMDP or HMPH protein protects against obesity and diabetes, and HMDP is also beneficial to cholesterol homeostasis.

## 1. Introduction

Diabetes is characterized by insulin resistance and/or insufficient secretion of insulin, hyperglycemia and hyperlipidemia, and type-2 diabetes mellitus (T2DM) accounts for over 90% of all diabetic patients [1]. In recent decades, the prevalence of T2DM has increased globally and more rapidly in the middle- and low-income countries (WHO, 2019). It was reported in 2018 that more than 500 million people were diagnosed to have T2DM worldwide [2]. It is projected that 642 million people, aged 20–79 years, will have diabetes by 2040 [3]. Diabetes causes numerous macrovascular and microvascular complications, and also increases the risk of developing other diseases, including cancer, ageing-related dementia, infection and liver disease [4]. Diabetes has become the seventh highest cause of death (WHO, 2019) and a serious worldwide threat to life quality, life expectancy and the health care system.

Despite the genetic susceptibility or predisposition [5], lifestyles including dietary habit and physical activity are closely related to the onset and progression of T2DM [6,7]. It is well-established that consumption of fat and carbohydrate is positively correlated with T2DM, while the effect of protein intake is inconsistent and depends on the type and origin of the protein ingested [8,9]. Several studies have shown that the consumption of red meat proteins is linked to the increase in T2DM cases while plant-based proteins are beneficial [10,11]. Although not clear, a high consumption of red meat proteins results in a concomitant high intake of lipids, in particular the saturated fatty acids, which have widely been recognized to increase the risk of developing diabetes and other metabolic disorders [12]. Nevertheless, fish proteins, such as those derived from cod and sardine, are reported to improve glucose metabolism and homeostasis [9,13,14]. Other nutrients such as polyunsaturated fatty acids (PUFA), antioxidants, peptides/hydrolysates of fish protein are also proven to be beneficial to the prevention and treatment of diabetes [15,16,17].

Fish milt has been treated historically as a waste during fish processing or used as a low-grade by-product. A recent report indicates that fish milt is rich in high quality proteins and lipids, particularly a high content of omega-3 fatty acids, eicosapentaenoic acid (EPA) and docosahexaenoic acid (DHA), which are predominantly in phospholipids [18]. Further, it is reported that fatty acids in the phospholipid fraction have a higher bioavailability and bioefficacy [19]. Due to high contents of protein, omega-3 fatty acids and other nutrients and bioactives, fish milt has been used as a food additive for malnourished children in some developing countries [20]. With a rapid increase in protein consumption as a result of the increase in population and understanding the beneficial effects of protein relative to fat and carbohydrate, market is demanding for more high-quality protein products. Atlantic herring provides over 50% of the international market and, consequently, a large quantity of milt is generated every year from the fish processing. In 2019, we reported for the first time the beneficial effects of herring milt protein hydrolysate (HMPH; herring milt with proteins being hydrolyzed enzymatically) on glycemic homeostasis and a number of metabolic phenotypes related to insulin function and glucose metabolism [21]. A similar but much weaker effect was reported in a more recent study conducted in the same animal model, where the diet-induced obese mice were treated with different extracts of herring milt protein hydrolysate that contained various contents of protein/peptides, lipids, nuclei acids and astaxanthin [22]. Due to the additional costs and processes associated with HMPH preparation, it is important to determine whether herring milt without pre-hydrolysis of proteins exerts similar anti-diabetic effects. Accordingly, we dried and ground herring milt into a powder form (HMDP) and conducted another study in the same animal model [23] to compare with HMPH for their effects on glucose control and related metabolic phenotypes. Herein, we report that HMDP is equally effective with HMPH in improving insulin resistance and pancreas β-cell function and thus blood glucose homeostasis in mice with high-fat diet-induced obesity and insulin resistance.

## 2. Results

### 2.1. Chemical Composition of HMDP and HMPH

The analysis and chemical composition of HMPH were reported previously [21,24], and the same methods were used to analyze the composition of HMDP. It was revealed that HMDP contained 69% of crude protein, 11.2% of lipids, 13.8% of ash, and 6.0% of moisture, similar to HMPH that had 70.8% crude protein, 10.8% lipids, 12.7% ash, and 6.6% moisture. HMDP had 29.3%, 29.3%, and 41.4% of saturated, monounsaturated, and polyunsaturated fatty acids, respectively, and an omega-3 to omega-6 ratio of 20.0, with 12.8% and 21.7% being EPA and DHA. HMPH contained 42.0%, 34.7%, and 23.3% of saturated, monounsaturated, and polyunsaturated fatty acids, respectively. The PUFA fraction had an omega-3 to omega-6 ratio of 10.7, with 8.0% and 10.8% being EPA or DHA. Among amino acids, l-arginine was accounted for 324.2 mg in HMDP and 292.3 mg in HMPH per gram of total amino acids. Taurine was 33.2 mg in HMDP and 30.8 mg in HMPH per gram of total amino acids. HMDP and HMPH had high contents of omega-3 PUFA, l-arginine and taurine and, further, HMDP had a higher content of the each aforementioned component than HMPH. In addition, HMDP and HMPH were different in the structure of proteins where HMDP had intact proteins, whereas proteins in HMPH were hydrolyzed by the enzyme reaction of protamex and alcalase [21].

### 2.2. Body Weight and Food Intake

Food intake was lower (*p* < 0.001) in the high-fat diet control (HFC) group than in the low-fat diet control (LFC) group throughout the treatment period (Figure 1). In comparison to the HFC group, the food intake of mice fed a modified high-fat diet where 70% casein protein was replaced with the equivalent amount of protein derived from HMDP (HMDP70) or HMPH (HMPH70) was decreased (*p* < 0.05) in the first week, but recovered in the second week and remained similar or higher (the HMPH70 group in weeks 3 and 4, *p* < 0.05) in other weeks. The HMDP70 and HMPH70 groups had similar food intakes all the time except in week 4 where the food intake was higher (*p* < 0.05) in the HMPH70 than in the HMDP70 group. The lower food intake of every group in week 5 might be attributed to the stress associated with oral glucose tolerance test (OGTT) performed during that week. Body weight was higher (*p* < 0.0001) in the HFC group than in the LFC group before the treatment and remained higher every week post the treatment (Figure 2). At the beginning of treatment, the HMDP70 and HMPH70 groups had similar body weight relative to the HFC group. After a week of treatment, the HMDP70 and HMPH70 groups did not differ but both became lighter (*p* < 0.05) than the HFC group, and the effect was retained throughout the treatment period. The relative weight of visceral fat mass (the sum of epididymal and perirenal fat pads) to the body weight was higher (*p* < 0.05) in the HFC than in the LFC group at the end of treatment, and did not differ among the HMDP70, HMPH70, and HFC groups (Table 1). The relative liver weight was similar among all groups.

### 2.3. Blood Glucose

Semi-fasting blood glucose levels are presented in Figure 3 where higher (*p* < 0.0001) levels were seen at every time point in the HFC group compared to the LFC group. After 2 w of treatment, the HMPH70 lowered (*p* < 0.05) semi-fasting blood glucose compared to the HFC group while the difference between the HMDP70 and HFC did not reach the significant level, which might be a result of large variations observed in the HMDP70 group. In weeks 4, 6, and 8, significant reductions (*p* < 0.05) were found in either the HMDP70 or the HMPH70 group as compared with the HFC group. No difference was detected at every time point between the HMDP70 and HMPH70 groups.

### 2.4. Oral Glucose Tolerance

The first oral glucose tolerance test (OGTT) was carried out after 4 w of treatment (week 5). The results showed that the HFC group had impaired glucose tolerance as evidenced by higher (*p* < 0.05) blood glucose levels than the LFC group at every time point post the oral glucose loading (Figure 4A). The HMDP70 and HMPH70 had a similar effect and lowered (*p* < 0.05) blood glucose at each time point compared to the HFC group. The OGTT was repeated after 8 w (week 9) of treatment. Similar to the results of the first OGTT, the HFC group showed higher glucose levels than the LFC group before and at each time point post the oral administration of glucose (Figure 4B). The HMDP70 and HMPH70 showed similar effects and both lowered (*p* < 0.05) blood glucose level relative to the HFC group at each time point. When the effect was evaluated using the area under the curve (AUC), the HFC exhibited higher values in both week 5 (*p* < 0.0001) and week 9 (*p* < 0.05) than the LFC group (Figure 5). Compared to the HFC group, the HMDP70 group showed a lower AUC after 5 w (*p* = 0.0001) and 8 w (*p* = 0.0006), respectively. Similar effect was found for the HMPH70 group and the AUC of this group was lower than the HFC group in the first (*p* < 0.0001) and second (*p* < 0.0001) OGTT, respectively. The AUC was not different between the HMPH70 and HMDP70 groups in the first or second OGTT.

### 2.5. Insulin Tolerance

As shown in Figure 6A, blood glucose decreased after insulin injection in mice of each group, reached the lowest levels at 60 min and then started to recover. The blood glucose was higher (*p* < 0.05) in the HFC group at the beginning of the insulin tolerance test (ITT) and at each time point following the insulin injection and recovered much faster than in the LFC group. At time 0, the HMDP70 (*p* = 0.0049) and HMPH70 (*p* = 0.0006) groups had lower blood glucose levels than the HFC group. The difference disappeared at 30 and 60 min in both groups relative to the HFC group. Blood glucose levels in the HMDP70 and HMPH70 groups did not recover as fast as in the HFC group and became lower (*p* < 0.01) at 90 and 120 min, respectively, than the HFC group. The AUC of ITT was 76% higher (*p* < 0.0001) in the HFC group than in the LFC group (Figure 6B). Compared with the HFC group, the AUC was lowered (*p* < 0.05) by 28% in the HMDP70 and by 32% in the HMPH70 group, respectively. The HMDP70 and HMPH70 groups did not differ in the blood glucose levels at each time point post the insulin injection and in the AUC.

### 2.6. Fasting Blood Glucose, Insulin, Leptin, and Free Fatty Acids

Fasting (12 h) blood glucose (FBG) was higher (*p* = 0.0024) in the HFC group than in the LFC group, and consistently demonstrated the impairment of glucose homeostasis in mice of the HFC group (Table 1). FBG was lowered in the HMDP70 (*p* = 0.0004) and HMPH70 (*p* = 0.0245) groups, respectively, compared to the HFC group. Fasting blood insulin was over 6-fold higher (*p* < 0.0001) in the HFC group than in the LFC group, and partially reversed in either HMDP70 (*p* = 0.0107) or HMPH70 (*p* = 0.0471) group. Similar effects were observed in the fasting blood leptin levels. The HFC group had an over 3-fold higher (*p* < 0.0001) fasting blood leptin levels than the LFC group, while a significant reduction was observed in both the HMDP70 (*p* < 0.0001) and HMPH70 (*p* = 0.0003) groups. Fasting blood free fatty acids were higher (*p* < 0.05) in the HFC than the LFC group and decreased in the HMDP70 (*p* = 0.0426) or HMPH70 (*p* = 0.0010) group. The HMDP70 and HMPH70 groups showed similar effects on any of these blood biomarkers.

### 2.7. Insulin Resistance and β-Cell Function Indices

The homeostasis model assessment of insulin resistance (HOMA-IR) index is presented in Table 1. It was over 7-fold higher (*p* < 0.0001) in the HFC group than in the LFC group. There was a significant treatment effect, with a 65% reduction (*p* = 0.0016) in the HMDP70 and a 49% reduction (*p* = 0.0216) in the HMPH70 group being observed compared to the HFC group. A marked difference (*p* < 0.0001) in the homeostasis model assessment of β-cell function (HOMA-β) was found between the HFC and LFC groups, with the HOMA-β index being 94% lower (*p* < 0.0001) in the HFC group relative to the LFC group. However, it was increased by 155% in the HMDP70 (*p* = 0.0012) group and by 82% in the HMPH70 (*p* = 0.0610) group. Quantitative insulin sensitivity check index (QUICKI) is another parameter used to assess the pancreas β-cell function and was lower (*p* < 0.0001) in the HFC group than the LFC group, in agreement with the HOMA-β index. A significant improvement of QUICKI was noticed in both the HMDP70 (*p* = 0.0013) and HMPH70 (*p* = 0.0277) groups compared to the HFC group. No difference was found between the HMDP70 and HMPH70 groups in HOMA-IR, HOMA-β, or QUICKI index.

### 2.8. Fasting Blood Lipid Profile

Fasting blood total cholesterol (TC) was increased by 59% (*p* = 0.0021) in the HFC group compared to the LFC group (Table 1). The HMDP70 decreased (*p* = 0.0019) the blood total cholesterol by 32% while the HMPH70 did not have a significant effect although a 10% reduction was observed. It was surprising to note that blood total cholesterol was lower (*p* = 0.0275) in the HMDP70 group than in the HMPH70 group. The blood high-density lipoprotein cholesterol (HDL-C) was higher (*p* = 0.0086) in the HFC group as compared with the LFC group, and the treatments did not show a significant effect. The HFC group had a 90% higher (*p* < 0.0001) non-HDL-cholesterol than the LFC group, and a 49% (*p* = 0.0034) decrease was found in the HMDP70 group relative to the HFC group. The HMPH70 group showed a 19% reduction in non-HDL cholesterol, which, however, did not differ from the HFC group. The HMDP70 group did not differ (*p* = 0.0659) from the HMPH70 group in blood non-HDL-cholesterol levels. The blood triacylglycerol (TAG) levels were not different (*p* = 0.0832) between the HFC and LFC groups, and no treatment effect was detected (*p* = 0.0718).

## 3. Discussion

The primary focus of the present study was to determine the effect of HMDP and HMPH on blood glucose homeostasis and further on the metabolic phenotypes that are closely related to glucose metabolism in a mouse model with high-fat diet-induced obesity and insulin resistance. The results demonstrated that mice fed a high-fat diet showed apparent metabolic disorders that were characterized by the elevation of blood glucose, insulin, leptin and cholesterol levels, impaired oral glucose tolerance, pancreatic β-cell dysfunction, and insulin resistance, in accordance with the results of previous studies in mouse and other animal models [15,23]. Being consistent with our previous study in high-fat diet-induced obese- and insulin-resistant mice [21], the replacement of 70% dietary casein protein with HMPH protein significantly attenuated hyperglycemia as early as 2 weeks post the treatment, which was evidenced by the significant decrease in semi-fasting blood glucose levels. A similar benefit was shown by HMDP, which lowered significantly the semi-fasting blood glucose following 4 weeks of treatment. The glucose-lowering effects did not differ between HMDP and HMPH throughout the study period. Substantial improvements on oral glucose tolerance and insulin sensitivity were observed in mice on either the HMDP70 or the HMPH70 diet. The beneficial effects of HMDP and HMPH were further demonstrated by a number of glucose metabolism-related phenotypes, including fasting blood insulin, leptin, free fatty acids, and lipids as well as HOMA-IR, HOMA-β, and QUICKI indices.

High-fat diet induces obesity, hyperinsulinemia, hyperglycemia, and insulin resistance in mice [15,21,25] and, therefore, this model has been widely used in T2DM research [15,26]. In the current experiment, male C57BL/6J mice on the high-fat diet showed a significant increase in weight gain although consumed less amounts of food, suggesting that a higher energy intake relative to energy expenditure occurred compared to the low-fat controls [15,27]. In accordance with our previous study [21], feeding mice with the HMPH70 diet resulted in a significantly lower body weight relative to the HFC group. HMDP was a new form of herring milt ingredient prepared by drying and grinding the fresh material collected during herring processing. When 70% dietary casein protein was substituted with the equivalent amount of protein from HMDP, weight gain was significantly decreased as observed in mice on the HMPH70 diet. Mice on the HMDP70 diet reduced food intake in the first week but had no effect in the remaining treatment period although slight and insignificant increases were noted in some of weeks as compared with the HFC group. Similarly, mice on the HMPH70 diet reduced food intake in the first week but increased food intake in weeks 3 and 4, while having no effect in week 2 and from week 5 to the end of the study. The increase in food intake in the HMPH70 group in weeks 3 and 4 was likely a result of compensation for the decrease in food intake in the first week. Our previous study showed that HMPH70 did not affect food intake in any weeks of the treatment compared to the HFC group [21]. As the food intake was not affected or tended to be higher in the HMPH70 and HMDP70 groups than in the HFC group except in the first week, the consistent reductions in weight gain in the HMPH70 and HMDP70 groups might be a result of decrease in energy absorption and utilization and/or increase in energy expenditure. Although not measured in the present study, several nutrients and functional ingredients high in HMDP and HMPH are reported to influence favorably the energy metabolism and balance. For example, dietary supplementation of herring milt enhances the hepatic fatty acid beta-oxidation and reduces fatty acid synthesis in mice [20]. Fish protein decreases fat absorption [28] and reduces appetite and increases basal energy expenditure in rats [29]. Inclusion of fish protein hydrolysate in mouse diet increases hepatic fatty acid oxidation and reduces body fat mass [30]. Apart from protein, HMDP and HMPH were rich in omega-3 EPA and DHA that are well-known for their benefits in increasing metabolic rate and lipid oxidation, resulting in the reductions in energy and fat storages [31,32]. In addition, substantial differences were found in the composition of each amino acid between casein and HMDP or HMPH (see Appendix A). The biggest differences existed in the content of l-arginine and taurine. HMDP and HMPH contained approximately 8–9-fold higher amount of l-arginine than casein. HMDP and HMPH also had significant amounts (over 30 mg/g total amino acids) of taurine while casein was taurine free. Evidence is emerging that l-arginine activates AMP kinase (AMPK), which is a metabolic sensor of energy metabolism at the cellular and whole-body level. Activation of AMPK leads to the increase in cellular catabolic capacity, fat and carbohydrate oxidations, and energy expenditure [33]. The beneficial effects of l-arginine supplementation on energy expenditure are consistent and independent of age, physical activity, and previous health status [34], and have been discussed extensively in a systematic review [35]. Taurine, another high content of amino acid in HMDP and HMPH, plays a vital role in energy metabolism, and its supplementation stimulates energy metabolism in muscle, cardiac, liver and adipose tissues [36,37].

Overnight fasting ruins diet-induced obese- and insulin-resistant model in rodents [21], therefore, semi-fasting (4–6 h) is used to monitor blood glucose levels during the treatment, and fasting blood glucose is measured only at the end of the experiment [15,25]. In line with weight changes, semi-fasting blood glucose was kept significantly higher in mice on the HFC diet than in those on the LFC diet during the entire treatment period. However, when mice were switched from the high-fat diet to the HMDP70 or HMPH70 diet, semi-fasting blood glucose was quickly lowered and remained low until the end of the study. A similar pattern was noted in the fasting blood glucose levels, which were higher in the HFC than in the LFC group and decreased in the HMDP70 and HMPH70 groups, respectively. There was a significant correlation between the body weight and fasting blood glucose (*r* = 0.6131, *p* < 0.0001), implying that blood glucose reductions in mice on the HMDP70 or HMPH70 diet might be a confounded result of the weight reduction and other beneficial effects induced by the active components in HMDP and HMPH, in agreement with our previous study [21]. Supportive observations are reported for other marine proteins such as cod–scallop, which decreases weight gain and serum glucose levels in diet-induced obese mice [38]. The benefits of fish protein to body weight, body composition, and blood glucose homeostasis have been confirmed in a recent clinical trial in insulin-resistant patients [39].

Elevation of semi-fasting and fasting blood glucose levels is indicative of insulin resistance, which is further evaluated by performing oral glucose tolerance and insulin tolerance tests and analyzing a body of blood biomarkers [15,25,40]. In the present study, mice of the HFC group exhibited significantly higher blood glucose levels at every time point in the two OGTTs as compared with mice of the LFC group, demonstrating the impairment of glucose tolerance in mice on the HFC diet, in accordance with the results of many previous studies [21,25,41,42]. Interestingly, mice fed the HMPH70 and HMDP70 diets displayed consistent reductions in blood glucose level at every time point during each OGTT and significantly lower AUCs as compared to mice fed the high-fat diet, with similar effects being noted for the two treatment groups. This notion was further supported by the results of ITT where a strong and long-lasting effect of insulin on glucose homeostasis was observed in both the HMDP70 and HMPH70 groups compared to the HFC group. The hypoglycemic effect of both products observed in this study was much stronger than the reported in a more recent study by Durand et al. in the same animal model [22]. The discrepancy might be a result of differences in the administration route, dose and composition of the test materials between the two studies. In our study, the entire herring milt was directly dried up or pretreated with enzymatic hydrolysis of proteins prior to dry up, whereas the study by Durand et al. used herring milt protein hydrolysate extracts that contained different protein, lipid, nuclei acid, and astaxanthin contents [22]. The dose of the test materials in the study by Durand et al. was 208.8 mg/kg by daily gavage, which was around 60-fold lower than the dose provided through diet in the present study. Circulating insulin is one of primary blood phenotypes in assessing insulin function and its elevation is the hallmark of insulin resistance [25,43]. The marked lowering of fasting blood insulin by the HMDP70 and HMPH70 diets implies the improvement of insulin sensitivity. A farther support was the reduction in HOMA-IR index in the HMDP70 and HMPH70 groups compared to the HFC group. Being consistent with our results, several studies in humans and animals demonstrate that consumption of fish proteins improves insulin sensitivity and glucose homeostasis [9,14,44]. The improved insulin resistance by HMDP and HMPH might be attributed to the beneficial effects of multiple bioactive components contained in HMDP and HMPH. A mouse study showed that dietary supplementation of herring milt tended to lower the plasma level of several pro-inflammatory cytokines [20]. Several in vivo and in vitro studies have shown that proteins, protein hydrolysates, or peptides of sardine, salmon, and tuna lower pro-inflammatory cytokine release, inflammatory cell count [13,45,46,47], and the production of inflammatory mediator iNOS [22]. The anti-inflammatory benefits of herring milt and fish proteins/hydrolysates may be related at least in part to their high contents of omega-3 EPA and DHA, which are well-established for the anti-inflammatory properties [20,30]. Additionally, studies have shown that protein hydrolysates and bioactive peptides obtained from fish and other marine sources of protein possess antioxidant and chelating activities and reducing power [47,48,49]. It is well-accepted that inflammation and oxidative stress are risk factors for insulin resistance, diabetes and obesity [50]. Several in vitro studies have demonstrated the regulatory effects of l-arginine on the release of growth hormone, insulin, and insulin-like growth factor-1 [51,52,53]. Insulin and growth hormone are counter-regulatory hormones in terms of glucose and lipid metabolism, and they mutually regulate the secretion of each other and form a complex regulatory network [54]. The balance between growth hormone and insulin-like growth factor-1 determines insulin secretion and sensitivity [55]. Moreover, l-arginine induces the activation of the mitogen-activated protein kinase (MAPK), AMPK [22,34,56], and sucrose nonfermenting AMPK-related kinase (SNARK) [22], which play important roles in the physiological adjustment of insulin singling and sensitivity and the subsequent maintenance of circulating glucose at appropriate levels [57]. Indeed, a study in high-fat diet-induced obese rats demonstrated that dietary supplementation of cod protein improved insulin signaling by restoring the insulin-induced activation of phosphatidylinositol 3-kinase/Akt and GLUT4-translocaton [9].

Pancreatic β-cell function is critical to glucose hemostasis by releasing insulin into the blood stream in response to the circulating glucose levels and can be assessed by HOMA-β and QUICKI indices [58]. A decrease in HOMA-β or QUICKI index means the deterioration of pancreatic β-cell function. The marked increase in HOMA-β and QUICKI indices in mice on the HMDP70 or HMPH70 diet compared to those on the HFC diet suggests that HMDP and HMPH are protective to pancreatic β-cell mass and/or function. This benefit might be a result of collective contributions by multiple nutrients and bioactives [59]. It is reported that peptides derived from fish protein positively influence pathways that control the body composition, lipid profile, and glucose metabolism [60], which all affect the β-cell proliferation and function. Recently, two critical reviews provided comprehensively the beneficial effects of omega-3 PUFA on pancreatic β-cells and insulin action [61,62]. Omega-PUFA prevent and reverse high-fat diet-induced inflammation and insulin resistance in adipose tissue, increase glucose-stimulated insulin secretion, and decrease prostaglandin production, which in turn enhances the secretion of insulin [61]. In pancreatic islets, omega-3 PUFA regulate cell proliferation and apoptosis and preserve β-cell mass [62]. Apart from protein and omega-3 PUFA, taurine is reported to promote insulin release in pancreatic β-cells [63], reduce insulin and leptin resistances, and thus ameliorate hyperglycemia and dyslipidemia [64,65,66]. Moreover, l-arginine is reported to potentiate glucose-induced insulin secretion from pancreatic β-cells under normal and diabetic conditions [67], a result of membrane depolarization and the rise of cytoplasmic Ca^2+^ through protein kinase A- and C-sensitive mechanisms [68]. A further study shows that l-arginine and its metabolite l-ornithine stimulate insulin secretion via G proteins, particularly the isoform Gαi2 [69].

The role of leptin in energy expenditure has been highlighted through studies conducted in recent decades [70]. Leptin is made in the adipose tissue and functions as an afferent signal through a negative feedback loop to regulate energy homeostasis and fat storage. The leptin endocrine system offers a crucial evolutionary function by keeping the relative constancy of adipose tissue mass and, therefore, protects individuals from being too light or too heavy [71] through a counter-regulation on insulin function [72]. As such, obesity or overweight is developed when leptin resistance occurs [73]. Leptin increases energy expenditure and the systemic and brown adipose tissue glucose utilizations as well as independently lowers blood glucose levels [74]. A recent review has summarized the role of leptin signaling in various hypothalamic nuclei and its effects on the sympathetic nervous system to influence brown adipose thermogenesis [70]. In addition, leptin potentiates cholecystokinin-induced satiation, inhibits insulin secretion, suppresses insulin-induced lipogenesis by stimulating lipolysis, and modulates peripheral tissue and brain sensitivity to insulin action [72]. The increase in blood leptin levels in obese mice is indicative of leptin resistance and vice versa [75]. In agreement with previous studies [21,25], increased leptin levels occurred in mice fed the high-fat diet and were reversed by the HMDP70 and HMPH70 diets, respectively. Similarly, sardine protein has been reported to lower plasma leptin and glucose levels and to improve the impaired glucose tolerance in rats with fructose-induced insulin resistance and metabolic syndrome [13]. As l-arginine modulates leptin function [76], it was possible that HMDP and HMPH improved leptin resistance at least in part via the action of l-arginine and thereby enhanced glucose utilization, stimulated energy expenditure and subsequently decreased body weight. Taurine, another important amino acid in HMPH or HMDP, is reported to restore leptin signaling in neurons and improve leptin resistance [77,78]. The reduction in serum leptin levels by HMDP and HMPH might be a consequence of improved leptin resistance or a confounded effect of improving both leptin and insulin resistances. On the other hand, it is impossible to exclude the possibility that HMDP or HMPH improved energy metabolism and lowered weight gain, which resulted in the improvement of leptin and insulin resistances as a secondary effect.

A recent study in humans found that nine amino acids (phenylalanine, tryptophan, tyrosine, alanine, isoleucine, leucine, valine, aspartate, and glutamine) were associated with the decreases in insulin secretion and the elevation of fasting and postprandial glucose levels, and of which five amino acids (glutamine, aspartate, tyrosine, leucine, and phenylalanine) also induce insulin resistance [79]. For example, the plasma level of tryptophan is reported to be negatively associated with type-2 diabetes [80]. Tryptophan is metabolized predominantly by the kynurenine pathway, and several metabolites of the kynurenine pathway are reported to be diabetogenic [81]. Tryptophan metabolites inhibit both proinsulin synthesis and glucose- and leucine-induced insulin release from pancreatic β-cells [82]. Except for alanine, the content of the other eight amino acids was lower in HMDP or HMPH than in casein, potentially beneficial to insulin secretion and sensitivity. Another study suggests that glutamine, leucine, and arginine activate mTORC1 and a high activation of mTORC1 in the muscle inhibits insulin signaling and contributes to systemic insulin resistance [83]. Compared to casein, HMDP and HMPH had over 7-fold higher amount of arginine but contained approximately half the amount of glutamine and leucine. As such, it is difficult to speculate the collective effect of these three amino acids in HMPH or HMDP on mTOR activity relative to casein control.

The analysis of blood free fatty acids revealed no difference between the HFC and LFC groups; however, the HMPH and HMDP diets moderately, but significantly, decreased the blood free fatty acid levels compared to the HFC group. Obesity and diabetes are associated with adipose dysfunction, which increases lipolysis and circulating free fatty acids and promotes ectopic fat deposit [84]. Lipid accumulation in the liver and muscle is the main cause of insulin resistance and T2DM [85,86,87], which explains why most of the subjects with T2DM are overweight or obese [88]. Increase in adipose lipolysis induces lipid accumulation in the liver, which in turn attenuates insulin sensitivity and promotes liver gluconeogenesis, resulting in the elevation of fasting blood glucose in diabetic patients [89]. Adipose tissue also produces adiponectin and the level of adiponectin is a marker of the lipolytic rate of triacylglycerols in the adipose tissue [90]. Adiponectin regulates adipocyte lipid storage and prevents ectopic lipid accumulations [91]. Some studies demonstrated a reciprocal relationship between blood concentrations of adiponectin and free fatty acids [25], while others did not support this notion [15,64]. Our published study conducted in the same animal model showed that blood adiponectin levels were not altered by HMPH [21] and thus were not analyzed in the current study.

High-fat diet induced the elevation of blood TC, HDL-C, and non-HDL-C levels in mice. It is not uncommon to see in rodents the increase in blood HDL-C when TC is increased because in rodents a significant portion of circulating cholesterol is carried by HDL [92]. Elevations of blood TC and non-HDL-C are a common phenomenon in the insulin-resistant and/or diabetic subjects, contributing to the development of atherosclerosis and cardiovascular diseases [93]. The HMDP70 diet significantly lowered the blood concentration of TC and non-HDL-C in high-fat diet-induced obese mice. This observation is consistent with the cholesterol-lowering effect of protein hydrolysates from salmon flesh remnants in Wistar and genetically obese Zucker (fa/fa) rats [94]. The HMPH70 diet decreased blood total cholesterol by 10% and non-HDL cholesterol by 19%. However, the effects were not significant, which is contradictory to the finding of our previous study in the same animal model [21]. The discrepancy is considered likely a result of large variations noticed within groups in the present study. Regulation of cholesterol metabolism and homeostasis involves multiple pathways and processes [21,95]. It is reported that fish protein and fish protein hydrolysate decrease intestinal cholesterol absorption by increasing fecal cholesterol and bile acid excretions [94,96]. Acyl-CoA:cholesterol acyltransferase is an enzyme that is essential for converting free cholesterol to cholesterol ester in the enterocyte post absorption, a critical step for secreting cholesterol from the basolateral side of the enterocyte into the lymphatic system before entering into the blood stream. Fish protein hydrolysates are reported to reduce the activity of this enzyme [94]. Fish protein also regulates the expression of genes that modulates the reverse cholesterol transportation as well as the liver cholesterol clearance and catabolism [97]. Moreover, fish proteins have been reported to interfere with the solubilization of cholesterol in micelles, which carry cholesterol across the unstirred water layer to the epithelium in the small intestinal lumen and make it ready for absorption [96]. The inhibition of micellar solubility of cholesterol in the intestine and cholesterol esterification in the enterocyte lowers cholesterol absorption [98]. HMDP showed better effects of lowering blood total cholesterol and non-HDL cholesterol levels, which might be attributed to its higher contents of omega-3 EPA and DHA [99], l-arginine [100], and taurine [101]. HMPH and HMDP did not significantly affect blood TAG levels, which was in line with the previous study on the effect of HMPH in the same animal model [21].

In conclusion, the replacement of 70% dietary casein protein with the equivalent quantity of protein derived from HMDP or HMPH in mice with high-fat diet-induced obesity and insulin resistance significantly improved semi-fasting and fasting blood glucose levels, oral glucose tolerance, insulin sensitivity and pancreatic β-cell function, along with the reductions in blood insulin, leptin and free fatty acid levels. HMDP and HMPH were demonstrated to be promising natural agents with similar potentials for preventing or treating obesity, diabetes and metabolic complications. HMDP reduced blood total cholesterol and non-HDL-cholesterol levels, which are the additional benefits to insulin-resistant and diabetic conditions. As a significant weight reduction was observed in mice treated with HMDP or HMPH, it is impossible to conclude that the observed benefits of HMDP or HMPH were a result of weight reduction, improvement of insulin secretion and sensitivity, or perhaps a confounded effect of weight reduction and independent improvement on insulin synthesis/secretion and function. Further studies may be conducted to determine bioactive components in HMDP or HMPH, especially proteins, peptides, amino acids, nuclei acids and fatty acids, responsible for the alteration of biomarkers and phenotypes related to glucose/energy metabolism and homeostasis. It is also important to explore the underlying mechanisms by measuring blood/tissue inflammatory and oxidative stress biomarkers, energy metabolism using a calorimetry system, and the tissue expression of molecular targets in the insulin-signaling pathway and MAPK and AMPK pathways.

## 4. Materials and Methods

### 4.1. Preparation and Analysis of HMPH and HMDP

HMDP was prepared using Refractance Window drying technology [102,103]. The initial herring milt contained 72% moisture, with a whitish gray color. Larger solid particles in the initial material were blended down with a Waring (Torrington, CT, USA) or a Silverson (East Longmeadow, MA, USA) blender to achieve a smooth puree. Twelve kilograms of the puree was then applied evenly to the belt that was run at a speed of 5 mm/s. The circulating water temperature under belt was set at 98 °C, resulting in the product temperature on the belt ranging from 35 °C to 50 °C. The material was dried using a Pilot GW dryer (G3 Enterprises, Inc., Modesto, CA, USA) down to 2.5–5% moisture in less than 10–12 min. The final product was about 2 kg, with off-white color and fish aroma. The preparation of HMPH and analysis of HMDP and HMPH were performed as reported previously [21]. It should be clarified that HMPH is a product of herring milt that was treated with a mix of protamex and alcalase to hydrolyze proteins prior to dry up, without extra steps or processes being employed to remove or modify the content of other bioactive compounds [21].

### 4.2. Animals and Diets

Thirty-six male C57BL/6J mice fed a high-fat diet (60% energy from fat) starting at the age of 5 w and 12 male C57BL/6J mice fed a low-fat diet all the time were obtained from Jackson Laboratories (Bar Harbor, ME, USA) at the age of 10 w. They were housed individually in mouse cages after arrival, with a 12:12 h light and dark cycle and free access to water and the same high-fat (D12492) or low-fat diet (D1250K) (Research Diets Inc., New Brunswick, NJ, USA). After 2 w of adaptation, mice were weighed and measured for glucose from tail vein blood after a semi-fasting (4–6 h) using a Breeze glucometer (Bayer, Zürich, Switzerland). The mice on the high-fat diet were then randomized into three groups based on the body weight and blood glucose concertation. One group continued on the high-fat diet and was used as the HFC, and the other two groups were provided with the HMDP70 or HMPH70 diet. Fat, starch, and fiber contents in the HMDP70 or HMPH70 diet were adjusted by deducting the equivalent amount of each nutrient from lard, corn starch, and cellulose in the respective diets. The mice on a low-fat diet all the time were used as the LFC. The diet composition is provided in the Appendix A. Body weight was obtained weekly and food intake was recorded daily for three consecutive days a week. Semi-fasting blood glucose was measured every other week with a Breeze glucometer (Bayer, Zürich, Switzerland) in blood from the tail vein. OGTT was performed twice and ITT was conducted once during the experimental period. After 9 w of the treatment, mice were fasted overnight and anesthetized with inhalation of isoflurane (Pharmaceutical Partners of Canada Inc., Toronto, ON, Canada). Blood was collected by cardiac puncture into serum collection tubes. Serum was collected by centrifuging the blood tubes at 1500× *g* for 20 min and stored in cryogenic vials at −80 °C. The liver and visceral fat were dissected and weighed. The animal use and experimental procedures were approved by the Joint Animal Care and Research Ethics Committee of the National Research Council Canada and the University of Prince Edward Island (25 March 2015; ACC Protocol no. 13-019 and File no. 6005333). The study was carried out following the guidelines of the Canadian Council on Animal Care.

### 4.3. Oral Glucose Tolerance Test

Oral glucose tolerance was determined during weeks 5 and 9 of the treatment, respectively, by administrating glucose orally at a dose of 2 g/kg body weight following 4–6 h fasting. Blood glucose concentration was measured from tail vein blood at 0, 30, 60, 90, and 120 min using a Breeze glucometer. The AUC of glucose concentrations was calculated and used as a second parameter to assess oral glucose tolerance.

### 4.4. Insulin Tolerance Test

Insulin tolerance was tested during week 7 of the treatment. Briefly, mice were fasted for 4–6 h prior to the intraperitoneal injection of human insulin at a dose of 0.7 U/kg body weight (Novo Nordisk Canada Inc., Mississauga, ON, Canada). Blood glucose was measured at 0, 30, 60, 90, and 120 min as described for the OGTT. The AUC of glucose concentrations was calculated and used as an additional parameter to evaluate insulin sensitivity or resistance.

### 4.5. Analysis of Fasting Blood Insulin and Leptin

Fasting blood insulin and leptin levels were determined in serum using commercial mouse ELISA kits following the kits’ instructions. The kits were purchased from Crystal Chem (Downers Grove, IL, USA). The concentration was calculated against the corresponding standards, which were run in parallel with the samples, and presented as ng/mL.

### 4.6. Analysis of Fasting Blood Lipids and Glucose

Fasting blood glucose, TC, HDL-C, and TAG were quantified in serum using enzymatic methods on a Pointe-180 Chemistry Analyzer (Pointe Scientific In., Canton, MI, USA), with the reagents being obtained from the manufacturer of the analyzer. Free fatty acids were analyzed using a free fatty acid quantification kit from BioVision (Mountain View, CA, USA) according to the kit instructions. The standards of glucose, cholesterol, triacylglycerols, and free fatty acids were run together with the samples and used to calculate the serum concentration of each parameter. The results are presented as mmol/L.

### 4.7. Homeostasis Model Assessment of Insulin Resistance and β-Cell Function

The HOMA-IR index was calculated as (fasting serum insulin in µU/mL × fasting blood glucose in mmol)/L/22.5. The HOMA-β index was calculated as (insulin in µU/mL × 20)/(glucose in mmol/L−3.5). QUICKI was calculated as 1/[log(fasting insulin in µU/mL) + log (fasting glucose in mg/dL)].

### 4.8. Statistical Analysis

All data were analyzed using SAS software version 9.4 (SAS Institute, Cary, NC, USA). The difference between the HFC and LFC was determined using the Student’s *t*-test while the treatment effects were detected using 1-way ANOVA. Repeated measures t-test and repeated measures 1-way ANOVA were used for parameters that were measured more than once. When a significant treatment effect was found, the differences among the HFC, HMPH70 and HMDP70 groups were further determined using the least-squares means test. The significance level was set at *p* < 0.05. The results are presented as means ± SEM.

## Figures and Tables

**Figure 1 marinedrugs-18-00635-f001:**
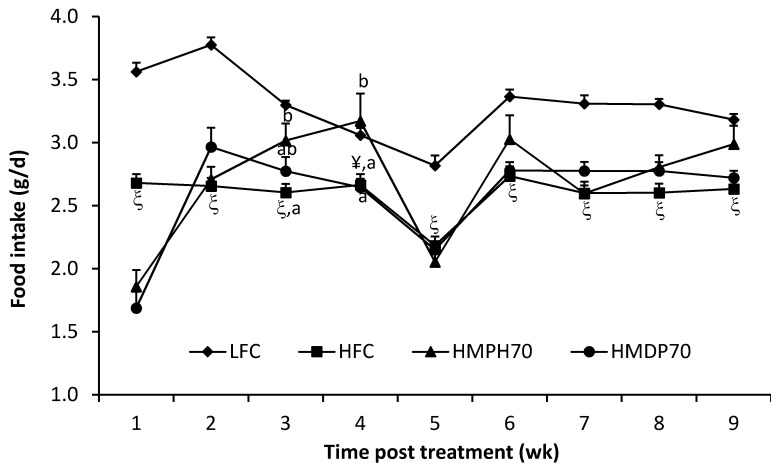
Effect of HMDP and HMPH on the food intake (g/d) of mice fed a high-fat diet. The results are means ± SEM (*n* = 10–12). The difference between the HFC and LFC groups was determined using repeated measures *t*-test. The treatment effect was analyzed using repeated measures 1-way ANOVA, and differences among the HFC, HMDP70 and HMPH70 groups were detected using the least-squares means test. The significance level was set at *p* < 0.05. ^¥^
*p* < 0.001, ^ξ^
*p* < 0.0001 compared to LFC. ^a,b^ Among HFC, HMDP70 and HMPH70 groups, values bearing different superscript letters differ, *p* < 0.05. LFC, low-fat control diet; HFC, high-fat control diet; HMDP70, HFC diet with 70% of casein protein being substituted with the same amount of protein derived from herring milt dry powder; HMPH70, HFC diet with 70% of casein protein being replaced with the equivalent quantity of protein derived from herring milt protein hydrolysate.

**Figure 2 marinedrugs-18-00635-f002:**
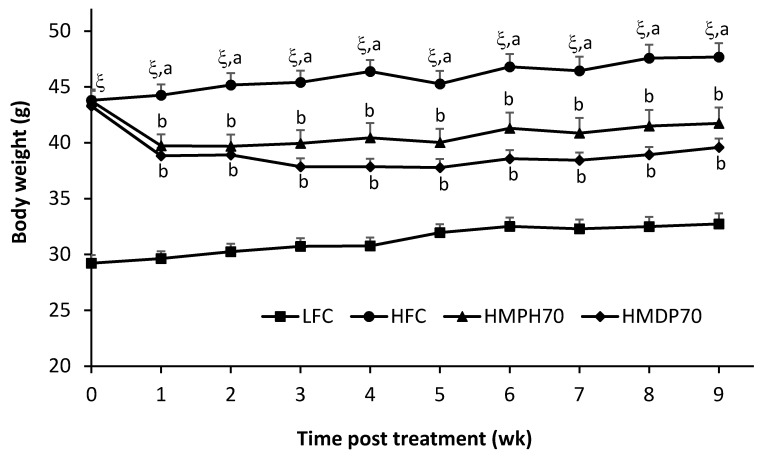
Effect of HMDP and HMPH on the body weight (g) of mice fed a high-fat diet. The results are presented as means ± SEM (*n* = 10–12). The difference between the HFC and LFC groups was analyzed using repeated measures *t*-test. The treatment effect was determined using 1-way ANOVA with repeated measures, and differences among the HFC, HMDP70 and HMPH70 groups were determined using the least-squares means test. The significance level was set to 0.05. ^ξ^
*p* < 0.0001 compared to LFC. ^a,b^ Among HFC, HMDP70 and HMPH70 groups, values bearing different superscript letters differ, *p* < 0.05. LFC, low-fat control diet; HFC, high-fat control diet; HMDP70, HFC diet with 70% of casein protein being replaced with the same amount of protein from herring milt dry powder; HMPH70, HFC diet with 70% of casein protein being substituted with the equivalent quantity of protein from herring milt protein hydrolysate.

**Figure 3 marinedrugs-18-00635-f003:**
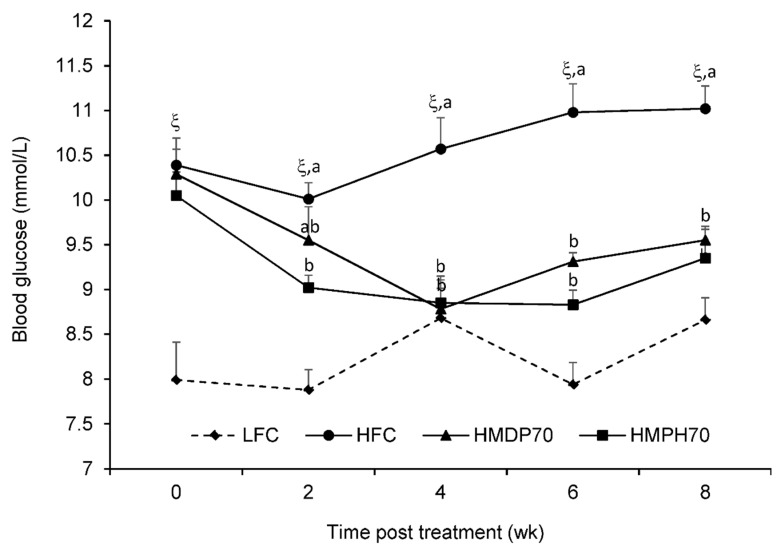
Effect of HMDP and HMPH on semi-fasting blood glucose in mice fed a high-fat diet. The difference between the HFC and LFC groups was analyzed using repeated measures *t*-test. The treatment effect was analyzed using 1-way ANOVA with repeated measures, and differences among the HFC, HMDP70 and HMPH70 groups were determined using the least-squares means test. The results are presented as means ± SEM (*n* = 11–12) and the significance level was set to 0.05. ^ξ^
*p* < 0.0001 compared to LFC. ^a,b^ Among the HFC, HMDP70 and HMPH70, values bearing different letters differ, *p* < 0.05. LFC, low-fat control diet; HFC, high-fat control diet; HMDP70, HFC diet with 70% of casein protein being replaced by the same amount of protein derived from herring milt dry powder; HMPH70, HFC diet with 70% of casein protein being replaced with the same amount of protein derived from herring milt protein hydrolysate.

**Figure 4 marinedrugs-18-00635-f004:**
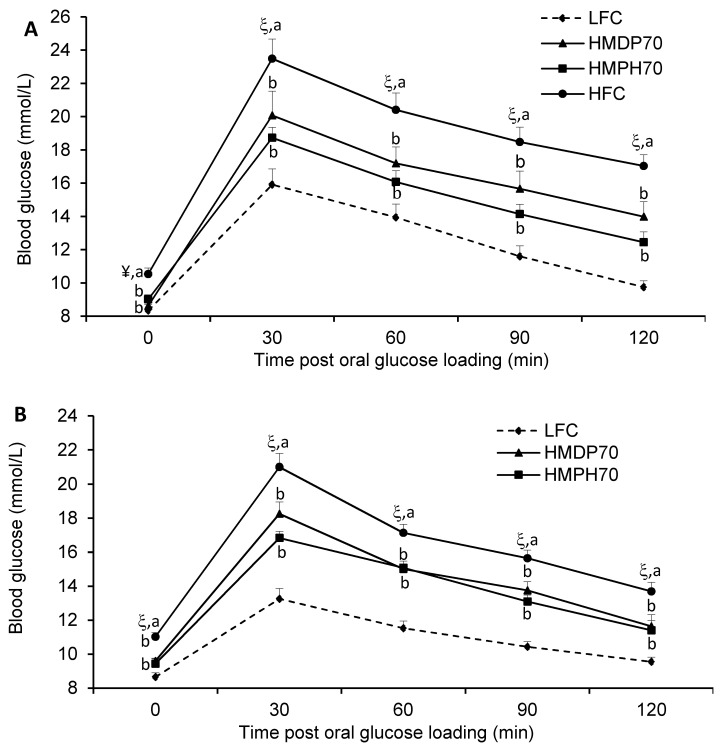
Effect of HMDP and HMPH on oral glucose tolerance in mice fed a high-fat diet. The difference between the HFC and LFC groups was analyzed using repeated measures *t*-test. The treatment effect was analyzed using repeated measures 1-way ANOVA, and differences among the HFC, HMDP70 and HMPH70 groups were determined using the least-squares means test. Data are presented as means ± SEM (*n* = 9–12). (**A**) The results of an oral glucose tolerance test conducted during week 5 of the treatment. (**B**) The results of an oral glucose tolerance test conducted during week 9 of the treatment. The significance level was set to 0.05. ^¥^
*p* < 0.001, ^ξ^
*p* < 0.0001 compared to LFC. ^a,b^ Among the HFC, HMDP70 and HMPH70 groups, values bearing different superscripts differ, *p* < 0.05. LFC, low-fat control diet; HFC, high-fat control diet; HMDP70, HFC diet with 70% of casein protein being replaced with the same amount of protein derived from herring milt dry powder; HMPH70, HFC diet with 70% of casein protein being substituted with the equivalent amount of protein from herring milt protein hydrolysate.

**Figure 5 marinedrugs-18-00635-f005:**
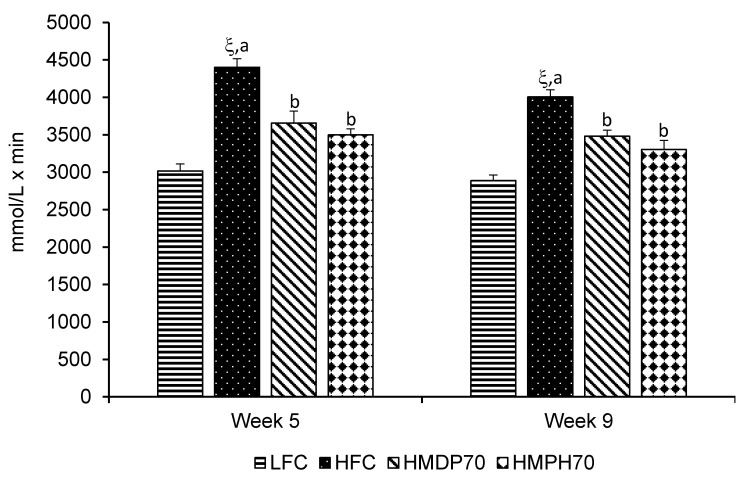
Effect of HMDP and HMPH on the area under the curve (AUC) of oral glucose tolerance in mice fed a high-fat diet. Data are means ± SEM (*n* = 9–12). The difference between the HFC and LFC groups was analyzed using Student’s *t*-test. The treatment effect was analyzed using 1-way ANOVA, and differences among the HFC, HMDP70 and HMPH70 groups were determined using the least-squares means test. The significance level was set to 0.05. ^ξ^
*p* < 0.0001 compared to LFC. ^a,b^ Among the HFC, HMDP70 and HMPH70 groups, values bearing different letters differ, *p* < 0.05. LFC, low-fat control diet; HFC, high-fat control diet; HMDP70, HFC diet with 70% of casein protein being substituted with the same amount of protein derived from herring milt dry powder; HMPH70, HFC diet with 70% of casein protein being substituted with the equivalent amount of protein from herring milt protein hydrolysate.

**Figure 6 marinedrugs-18-00635-f006:**
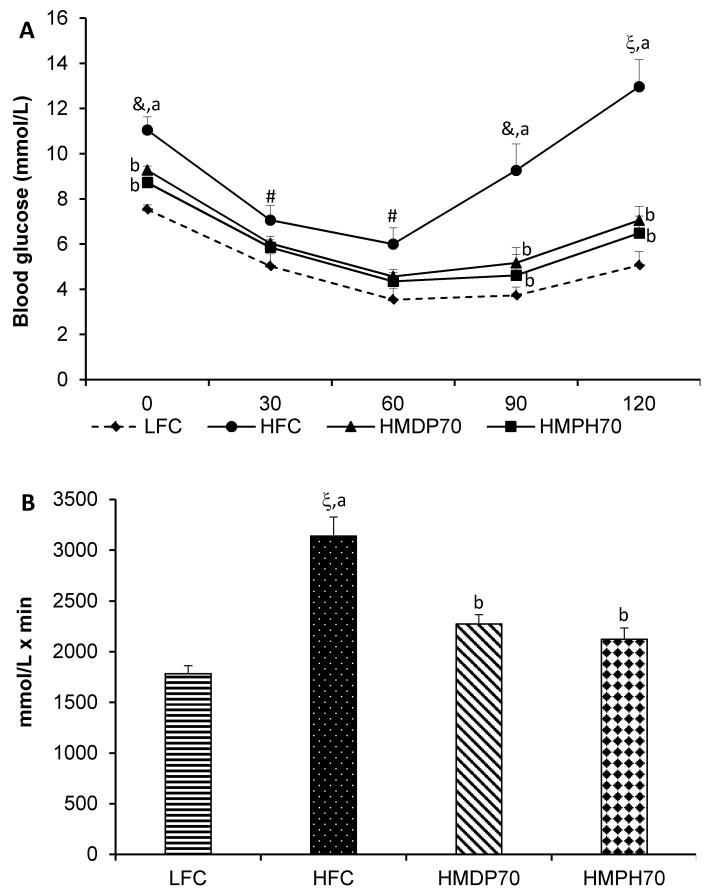
Effect of HMDP and HMPH on insulin tolerance during week 7 of the treatment in mice fed a high-fat diet. Data are presented as means ± SEM (*n* = 6). (**A**) The results of insulin tolerance test. (**B**) The area under the curve (AUC) of the insulin tolerance test. The difference between the HFC and LFC groups was analyzed using repeated measures *t*-test (**A**) or Student’s *t*-test (**B**). The treatment effect was analyzed using repeated measures 1-way ANOVA (**A**) or 1-way ANOVA (**B**), and differences among the HFC, HMDP70 and HMPH70 groups were determined using the least-squares means test. The significance level was set to 0.05. ^#^
*p* < 0.05, ^&^
*p* < 0.01, ^ξ^
*p* < 0.0001 compared to LFC. ^a,b^ Among the HFC, HMDP70 and HMPH70 groups, values bearing different letters differ, *p* < 0.05. LFC, low-fat control diet; HFC, high-fat control diet; HMDP70, HFC diet with 70% of casein protein being replaced by the same amount of protein from herring milt dry powder; HMPH70, HFC diet with 70% of casein protein being replaced by the same amount of protein from herring milt protein hydrolysate.

**Table 1 marinedrugs-18-00635-t001:** Effect of HMDP and HMPH on fasting blood glucose, insulin, leptin, lipids, β-cell function index, insulin resistance index and relative liver and visceral fat weights in mice fed a high-fat diet.

	LFC	HFC	HMDP70	HMPH70
FBG (mmol/L)	10.42 ± 0.68	14.25 ± 0.87 ^&,a^	10.94 ± 0.40 ^b^	12.18 ± 0.43 ^b^
Insulin (ng/mL)	0.13 ± 0.03	0.85 ± 0.10 ^ξ,a^	0.51 ± 0.10 ^b^	0.50 ± 0.09 ^b^
Leptin (ng/mL)	7.75 ± 1.41	23.81 ± 0.64 ^ξ,a^	16.10 ± 1.05 ^b^	18.16 ± 1.10 ^b^
HOMA-IR	1.55 ± 0.37	12.83 ± 1.76 ^ξ,a^	5.82 ± 1.17 ^b^	6.59 ± 1.20 ^b^
HOMA-β	1.83 ± 0.47	0.11 ± 0.02 ^ξ,a^	0.28 ± 0.05 ^b^	0.20 ± 0.04 ^ab^
QUICKI	0.38 ± 0.02	0.27 ± 0.01 ^ξ,a^	0.31 ± 0.01 ^b^	0.30 ± 0.01 ^b^
TC (mmol/L)	2.80 ± 0.24	4.45 ± 0.40 ^&,a^	3.03 ± 0.22 ^b^	4.01 ± 0.23 ^a^
HDL-C (mmol/L)	1.49 ± 0.08	1.94 ± 0.13 ^&^	1.75 ± 0.07	1.98 ± 0.05
Non-HDL-C (mmol/L)	1.32 ± 0.21	2.51 ± 0.37 ^&,a^	1.28 ± 0.21 ^b^	2.03 ± 0.23 ^ab^
TAG (mmol/L)	0.54 ± 0.06	0.71 ± 0.07	0.54 ± 0.06	0.53 ± 0.04
FFA (mmol/L)	0.50 ± 0.03	0.54 ± 0.01 ^a^	0.47 ± 0.02 ^b^	0.42 ± 0.03 ^b^
Visceral fat/body weight (%) ^1^	2.98 ± 0.29	4.60 ± 0.22 ^¥^	4.39 ± 0.12	4.46 ± 0.29
Liver/body weight (%)	2.94 ± 0.11	3.34 ± 0.32	3.42 ± 0.15	3.89 ± 0.28

^1^ Visceral fat was the sum of epididymal and perirenal fat pads. The weight of mesenteric fat pad was not obtained due to technical difficulty for an accurate dissection. The results are means ± SEM (*n* = 10–12). The difference between the HFC and LFC groups was determined using the Student’s *t*-test. The treatment effect was analyzed using 1-way ANOVA and differences among the HFC, HMDP70 and HMPH70 groups were determined using the least-squares means test. The significance level was set at *p* < 0.05. ^&^
*p* < 0.01, ^¥^
*p* < 0.001, and ^ξ^
*p* < 0.0001 compared to LFC. ^a,b^ Among the HFC, HMDP70 and HMPH70, values bearing different superscript letters are different, *p* < 0.05. FBG, fasting (12-h) blood glucose; FFA, free fatty acids; HDL-C, high-density lipoprotein cholesterol; TAG, triacylglycerols; TC, total cholesterol; HOMA-β, homeostasis model assessment of β-cell function; HOMA-IR, homeostasis model assessment of insulin resistance; QUICKI, quantitative insulin sensitivity check index; LFC, low-fat control diet; HFC, high-fat control diet; HMDP70, HFC diet with 70% of casein protein being replaced with the same amount of protein derived from herring milt dry powder; HMPH70, HFC diet with 70% of casein protein substituted with by the same amount of protein from herring milt protein hydrolysate.

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
