# Peer review of "Herring Milt and Herring Milt Protein Hydrolysate Are Equally Effective in Improving Insulin Sensitivity and Pancreatic Beta-Cell Function in Diet-Induced Obese- and Insulin-Resistant Mice"

_marinedrugs, 2020, doi:10.3390/md18120635_

Round 1
Reviewer 1 Report
The authors compared the anti-obese properties between milt protein dry powder (HMDP) and herring milt protein hydrolysate (HMPH) using high-fat diet induced obese mice. Both suppressed body weight increase and ameliorated related parameters except lipid metabolism.
Major comments.
Durand R et al. recently published a paper “Animal and Cellular Studies Demonstrate Some of the Beneficial Impacts of Herring Milt Hydrolysates on Obesity-Induced Glucose Intolerance and Inflammation (Nutrients. 2020)”. Compared to this Nutrients’ paper (IF=4.17), this manuscript does not reach the scientific depths for publication in Marine Drugs (IF=4.37).
Reviewer 2 Report
The authors report on the use of herring milt as an alternative protein source in a mouse model for type 2 diabetes. The authors have previously reported on herring milt hydrolysate (HMPH) and T2D, but the current study further expand on that by evaluating a far cheaper preparation method with dried herring milt (HMDP). For most effects that they evaluate, HMPH and HMDP are equivalent, which is quite important for large-scale implementation of this readily available by-product protein source, since HMDP most likely will be much cheaper. HMDP might even have additional benefits by lowering blood cholesterol (Table 3) more than HMPH.
The experiments done and the analyses done are sound and the text is easy to follow, but contains a few errors and could benefit from another read-through ( for example a trailing "with" after t-test in the Figure 1 legend).
My main criticism against the paper is that it is highly descriptive and does not attempt to characterize the mechanism of protection from herring milt "protein" in T2D. The authors correctly point out that this is a complex drug consisting of protein, long omega-3 unsaturated fatty acids, L-arginine and potentially other small molecules that may influence inflammation and metabolism. These potential effects are all discussed but no attempts are made to determine what or which of these components that play an important role in this preparation. The fact that the HMPH had similar effects as HMDP indicates that the effects are not due to protein activities (except perhaps the cholesterol-lowering effect), since hydrolysed product showed the same effect.
Something not discussed here was for example whether the amino acid composition of caseine and HM{PH,DP} are different. Catabolic products of different amino acids can form signaling molecules and hormones that influence many physiological aspects (for example, tryptophan can be a precursor for auxin/IAA, serotonin and other indole compounds, glutamine regulates mTOR activity in many cells, ...)
Also indirect physiological effects could be important, similar to how high-fiber diet promotes gut microbiota that generates short fatty acids, which in turn serve as energy source for anti-inflammatory immune cells. A lot of the tryptophan --> IAA conversion is also done by gut microbiota.
One follow-up study I would like to see is to treat mice with HMDP in presence of broad antibiotics to determine whether the protective effect in T2D is mediated via the microbiota or by direct molecular effects from HMDP components on the animal cells.
Author Response
Please see the attachement.

Reviewer 3 Report
marinedrugs-981822
Wang et al. shown how Herring milt and herring milt protein hydrolysate are able to improve insulin sensitivity in DIO mice.
The manuscript is well written, structured, discussed and the aim and the conclusions are clear.
However, there are important points that need to be clarified.
1.-To be more visual, Table 1 (food intake) and Table 2 (Body weight) should be presented in scatter plot.
2.- It is necessary to clarify which means “abdominal fat weights” or “visceral fat”. There are some fat depots considered abdominal (mesenteric, retroperitoneal, even gonadal). Thus, it is necessary to clarify this point.
3.- Please harmonize the error bars in the different figures. Some of them are shown on both sides and others just above (Figure 2A).
4.-It is necessary to show the insulin signalling in peripheral tissues (adipose tissue, muscle or liver) to conclude that the treatments ameliorate the insulin sensitivity. It can be done analysing phospho-AKT levels in the different treated groups, or the presence of glucose transporters in the plasmatic membrane
5.-It is necessary to show if the different treatments are able to reduce the plasma inflammatory parameters such as TNFa.
6.-Calorimetry experiments are necessary to check if the treatments promote a better energy consumption.
Round 2
Reviewer 1 Report
Dear authors,
Thank you for the explanation of the difference between your study and Durand R, et al.'s one.
But it's unlucky to you, they already published the similar story with Herring Milt Hydrolysates's effects againt obesity.
That means, you have to shown and prove the therapeutic mechanisms of Herring Milt Hydrolysates, when you try to get accepatance by the same level journal. I think Nutrients and Marine Drugs are the same level.
It's a pity for you, but sometimes happens in our field.
Reviewer 3 Report
The authors have improved the manuscript, but there are important questions that must be resolved before publishing. It is not enough to introduce them as limitations, because in this case the limitations are too important.
Round 3
Reviewer 3 Report
Excellent work. Congratulations.